# Amphipathic Solvent-Assisted Synthetic Strategy for Random Lamellae of the Clinoptilolites with Flower-like Morphology and Thinner Nanosheet for Adsorption and Separation of CO_2_ and CH_4_

**DOI:** 10.3390/nano13131942

**Published:** 2023-06-26

**Authors:** Jiawei Zhou, Bingying Jia, Bang Xu, Jihong Sun, Shiyang Bai

**Affiliations:** Beijing Key Laboratory for Green Catalysis and Separation, Department of Chemical Engineering, Beijing University of Technology, Beijing 100021, China

**Keywords:** clinoptilolites, random lamellae, fractal, o-Phenylenediamine, CH_4_/CO_2_ separation

## Abstract

The random lamellae of the synthetic CP were synthesized with a hydrothermal approach using o-Phenylenediamine (OPD) as a modifier. The decreases in the order degree of the CP synthesized in the presence of the OPD resulted from the loss of long-range order in a certain direction. Subsequently, the ultrasonic treatment and washing were conducive to further facilitate the disordered arrangements of its lamellae. The possible promotion mechanism regarding the nucleation and growth behaviors of the sol-gel particles was proposed. The fractal evolutions of the aluminosilicate species with crystallization time implied that the aluminosilicate species became gradually smooth to rough during the crystallization procedures since the amorphous structures transformed into flower-like morphologies. Their gas adsorption and separation performances indicated that the adsorption capacity of CO_2_ at 273 K reached up to 2.14 mmol·g^−1^ at 1 bar, and the selective factor (CO_2_/CH_4_) up to 3.4, much higher than that of the CPs synthesized without additive OPD. The breakthrough experiments displayed a longer breakthrough time and enhancement of CO_2_ uptake, showing better performance for CO_2_/CH_4_ separation. The cycling test further highlighted their efficiency for CO_2_/CH_4_ separation.

## 1. Introduction

Natural gas has been one of the most important fossil fuels for nearly two decades, which has been widely used in the fields of electricity generation and industrial applications due to its advantages of being clean and cheap [1,2,3]. However, the ever-growing demands exacerbate its desperate shortage, and thus an improvement in utilization efficiency and the recovery of waste CH_4_ become an urgent task for both industry and academia. Among the numerous strategies, the purification of biogas for CH_4_ enrichment is considered a promising solution [4,5,6,7]. Meanwhile, its application is also increasing as a means of reducing the pollution effects of organic biomass [8,9,10], although it is generally considered a low-grade natural gas due to the main components of CH_4_ (55–65%) and CO_2_ (35–45%) [11]. In these regards, increasing its calorific value by reducing the content of CO_2_ is essential for its efficient utilization [12].

Currently, the main technologies have been explored for CO_2_/CH_4_ separation, including cryogenic separation [13,14], membrane separation [15,16,17], water scrubbing [18,19], organic solvent scrubbing [20,21] and solid adsorbent adsorption separation [22,23,24]. The solid adsorbents, such as porous zeolites, carbon molecular sieves, titanium silicate, metal–organic frameworks, activated carbon and silica gel [25,26,27], presented many advantages of great separation efficiency and renewable capability and cost-effectiveness [28]. Generally, zeolites are more suitable for adsorbing acid gas, due to their distinctive microporous structure, excellent ion exchange behaviors and high adsorption ability [29]. For example, Maheshwari et al. [30] reported that an approach using expanded MCM-22(P) at high pH and room temperature obtained increased layer spacings without disrupting the framework connectivity. Furthermore, the expanded MCM-22(P) can recover its original structure with a characteristic layer spacing of 2.7 nm after acidification at a relatively lower temperature. However, the reversibility was lost at above 55 °C because swelling procedures at high temperatures easily resulted in partial dissolutions of framework silica and destruction of the layered structures. Shetti et al. [31] found that the catalytic behaviors of the monolamellar and multilamellar MFI nanosheets were greater than those of conventional MFI zeolite. The improved catalytic activities can be assigned to a shorter molecular diffusion path and a great number of acid sites that appeared on the mesopore surfaces of the MFI nanosheets. Obviously, the diffusion efficiency of the guest molecules could be improved by reducing the thickness of the MFI zeolite crystals on the mesoporous and microporous length scales. Runnebaum et al. [32] further demonstrated that exfoliated zeolite UCB-3 exhibited a 3.5-fold superior catalytic action about its three-dimensional Al-SSZ-70 zeolite for the alkylation of toluene with propylene at 423 K, and thereafter proposed that the exfoliated procedure of 3D Al-SSZ-70 zeolite was beneficial to promote the zeolite–catalyzed reaction rates, due to comparative increase in external surface area and acid sites in a certain direction. Although Corma et al. [33] successfully synthesized ITQ-2 with an MWW-type structure using MCM-22(P) as a precursor at 353 K using sonication and pickling, the high temperatures and strong acidifications easily led to the skeleton desilication. Choi et al. [34] prepared swollen derivatives of AMH-3 with intercalation of primary amine molecules (dodecylamine) at room temperature. Eilertsen et al. [35] proposed that the layered zeolite PREFER could be exfoliated using a mixed solution of cetyltrimethylammonium bromide, tetrabutylammonium fluoride and tetrabutylammonium chloride. However, undesirable heteroatom leaching from the framework occurred after HCl-acidification at room temperature. Similarly, Ogino and co-workers obtained a 1.5-fold increase in the quantity density of the external surface area and acid sites of SSZ-70 after treatment with a surfactant followed by sonication. The absence needed for acidification in the synthesis process can cause the exfoliation of heteroatom-containing SSZ-70 while preserving the skeletal integrity of heteroatoms [36].

Clinoptilolite (CP), a member of the heulandite (HEU) group, is one of the most abundant natural zeolites. Its structure consists of two parallel 10-membered ring channels A and B (0.44–0.72 nm and 0.40–0.55 nm, respectively), as well as a linked 8–membered ring channel C (0.41 × 0.40 nm), which intersects A and B [37,38]. The unique micropore structure selectively separates CO_2_/CH_4_ gas mixtures because of the different kinetic diameters of CO_2_ (0.34 nm) and CH_4_ (0.38 nm). In addition, two-dimensional zeolite with a high ratio of lateral size to thickness can be obtained using the swollen and exfoliation method, because its crystalline is weakly assembled in a particular direction.

Pour et al. [39] investigated the adsorption performances of CP for CO_2_ removal from CO_2_/CH_4_ mixtures at 277–310 K for pressures reaching 10 bar. The CP-adsorbed outcomes improved with pressure but decreased with temperature, reaching 2.80 mmol/g of CO_2_ and 1.39 mmol/g of CH_4_ at 277 K and 10 bar, respectively. The direct correlation between temperature and the kinetic energy of molecules at a lower temperature explains the higher adsorption capacity. Furthermore, the selectivity of CO_2_/CH_4_ dropped to a constant value with increasing pressure, showing around 5.63 at 277 K and 1 bar. This is mainly due to the high CO_2_ adsorption tendency of the adsorbents in the low-pressure region and the existence of saturated adsorption sites on the surface of the zeolite. Kennedy and co-workers [40] discussed the adsorption equilibria and the diffusion rates at 303 K for up to 8 bar using metal cation (Li^+^, Ca^+^, Cs^+^, Ni^2+^ and so on)-exchanged natural CP as the adsorbents. These screening analyses showed that the Cs-CP had a greater performance for CO_2_/N_2_ and CH_4_/N_2_ equilibrium separations. However, both Li- and Ni-CPs presented a favorable kinetic selectivity for N_2_/CH_4_ separations, the Ca-CP displayed superior selectivity for CO_2_/CH_4_ and CH_4_/N_2_ equilibrium separations. The probable reason was owing to pore blocking reducing CH_4_ equilibrium adsorbed capacity and thereafter increasing microporous diffusion resistance.

Recently, Wang et al. [38] proposed a method for producing CP using alcohol solvents with different morphologies (spherical, flower, cylindrical and disc-shaped), and further explored the selectivity and capacity of their CH_4_ and N_2_ adsorption. The flower-like CP presented an ideal selectivity for CH_4_/N_2_ of about 5.58 at 273 K and 1 bar, much higher than conventional CP (only 1.21). Obviously, the morphologies of the obtained CPs may strongly affect the CH_4_/N_2_/CO_2_ adsorption capacity and selectivity via modulating surface characteristics and adjusting diffusion properties of guest molecules, which is of great significance for their possible applications in the adsorption and separations of the mixed gases.

In this work, the random lamellae of the synthetic CP were prepared using amine-assisted ultrasonication. The synthetic CPs were firstly modified with an amphiphilic molecule (o-Phenylenediamine, abbreviated as OPD). Subsequently, ultrasonication (U) and ethanol washing (E) were performed to obtain the random lamellae of the synthetic CP (named CP-n-U-E, where n represents the molar percentage of OPD with SiO_2_). Particularly, the random lamellae of the synthetic CP could be achieved at mild conditions without acidification, and, therefore, the desilication phenomena could be avoided, which is one of the significant differences from the current literature reports [33,35]. Meanwhile, the activation energies of the induction period (*En*) and growth procedure (*Eg*) of the CP synthesized with additive OPD were speculated. Specifically, the fractal structural evolutions during the crystallization stages of the synthesized CP with additive OPD, sonication and ethanol washing were elucidated with small-angle X-ray scattering (SAXS) patterns combined with other characterizations, such as X-ray diffraction (XRD), Fourier transform infrared (FT-IR) spectra, thermogravimetric (TG) profiles, N_2_ sorption isotherms, the scanning electron microscope (SEM), the transmission electron microscope (TEM) and ^29^Si-nuclear magnetic resonances (NMR) profiles. The modified effect originating from the introduction of an amphiphilic molecule (OPD) was proposed, which played a significant role in decreasing surface energy during CPs crystallization and the formation of the random lamellae of the synthetic CP. Finally, the obtained CP-n-U-E was used as an adsorbent to evaluate its CH_4_ and CO_2_ adsorption capacities and separation. Their structural stability and cycling behaviors were preliminarily explored.

## 2. Materials and Methods

### 2.1. Materials

The aqueous colloidal silica sol (30 wt% SiO_2_) was provided by Qingdao Ocean Chemical Co., Ltd. (Zhongshang, China), CHN. Al(OH)_3_ (99.5 wt%) was purchased from Tianjin Fuchen Chemical reagents Co., Ltd. (Tianjin, China), CHN. NaOH (96.0 wt%) and KOH (82.0 wt%) were obtained from Tianjin Guangfu technology development Co., Ltd., CHN. Ethanol (99.7 wt%) was provided by Tianjin Damao chemical reagent Co., Ltd. (Tianjin, China), CHN. OPD was supplied by Shanghai Macklin Biochemical Technology Co., Ltd. (Shanghai, China), CHN. The used materials were all of the analytical reagent grades. As a seed, natural CP was sieved using a 400-mesh sieve. All experiments used deionized water with a resistivity of 18.25 MΩ∙cm.

### 2.2. Synthesis of Synthesized CP

The synthesis of CP followed the procedures described in the literature [38]. First, to prepare a clear solution, NaOH, KOH, Al(OH)_3_, OPD and deionized water were mixed and agitated at 150 °C for 3 h. Then, deionized water and silica sol were added slowly dropwise to the clear solution. The molar ratios of the above components were as follows 0.12 Na_2_O: 0.12 K_2_O: SiO_2_: 0.09 Al_2_O_3_: 26.27 H_2_O: n OPD. After that, the natural CP was poured into the mentioned above mixture and stirred at 25 °C for two hours. Then, the mixed solution was poured into a Teflon vessel in a steel autoclave and hydrothermally treated for 72 h at 150 °C. After cooling to 25 °C, it was filtered and washed with deionized water and dried for 12 h at 120 °C. The obtained solid was named synthesized CP.

### 2.3. Random Lamellae of Synthesized CP

A total of 0.2 g of synthesized CP was added into 30 mL of deionized water followed by ultrasonic treatment at a frequency of 50 kHz for 6 h. The solid was obtained after centrifuging and drying at 120 °C for 6 h and named CP-n-U. After that, the dried solid was mixed with 10 mL of ethanol and stirred at room temperature for 2 h. Finally, the obtained solid was dried at 120 °C for 3 h and named CP-n-U-E.

### 2.4. Characterizations

Cu K radiation (λ = 0.154056 nm) was used to create the XRD patterns with the XD-6 X-ray diffractometer (Beijing Purkinje General Instrument Co. Ltd. (Beijing, China), CHN.) in the 2*θ* range of 5–50° at 36 kV and 20 mA. SEM (JEM-2100F, JEOL Ltd. (Tokyo, Japan)) and TEM (JEOL-2010, JEOL Ltd.) images, taken with microscopes operating at 15.0 and 200 kV, respectively, were used to investigate the morphologies and microstructures of the samples. A 15 mg sample was used to measure the TG profiles with a PerkinElmer Pyris I thermal analyzer (Waltham, MA, USA). The experiments were carried out in an N_2_ atmosphere at a gas flow rate of 20 mL min^−1^ and a heating rate of 5 °C min^−1^ from 30 to 900 °C. The FT-IR spectrum was measured with an IR Prestige-21 FT-IR spectrophotometer (Shimadzu, (Tokyo, Japan)) using KBr as a medium in the 400–4000 cm^−1^ wavenumber range. JWGB JW-bk300 from Beijing Sci. & Tech. Co. Ltd. (Beijing, China), CHN. was used to investigate the adsorption–desorption isotherms of N_2_. At a liquid nitrogen temperature of 77 K, N_2_ adsorption–desorption isotherms were determined after all samples had been degassed for 6 h under a high vacuum at 120 °C. The surface area was estimated using the BET equation. The mesopore size distribution was calculated using the desorption branch of the isotherms based on the BJH model. The ^29^Si NMR patterns were performed using an Agilent 600M solid nuclear magnetic resonance spectrometer (Agilent Technologies Inc. (Palo Alto, CA, USA)). The Si/Al molar ratio was calculated on the basis of ^29^Si NMR patterns with a resonance frequency of 119.20 MHz and MAS of 12 kHz and calculated with Equation (1):(1)$$Si/Al=\frac{\sum_{n=0}^{4}A_{Si\left(nM\right)}}{\sum_{n=0}^{4}0.25nA_{Si\left(nM\right)}}$$
where *M* represents Al, *A* is the fitted NMR spectrum peak area, and *n* is Si(*n*Al).

The SAXS pattern was studied at the 1W2A station of the Beijing Synchrotron Radiation Facility using synchrotron radiation as the X-ray source and an incidence X-ray wavelength of 0.154 nm [41,42]. The 1600 mm sample-to-detector distance was matched using the standard sample’s diffraction ring as a reference. Before taking sample measurements, a mask measure of the detector readout noise (dark current) was completed. Mar165 CCD revealed that it was approximately 10 counts per second. A sample cell was filled with the 1 mm thick sample, and a groove was sealed using Scotch tape. The scattering photos were gathered using a 5 min exposure duration and a single-frame mode with a 2 time “multi-read”. The two-dimensional SAXS pictures were converted into one-dimensional data using the *Fit2D* application (http://www.esrf.eu/computing/scientific/FIT2D, accessed on 1 December 2022) before being further processed using the *S* software suite [43].

### 2.5. Adsorption Measurements

#### 2.5.1. Adsorption Volume Method

The CH_4_ or CO_2_ sorptions were measured with a JWGB JW-BK300 (Tokyo, Japan) gas sorption analyzer. Before measurements, the samples were activated at 120 °C under a vacuum for 6 h. Then, the equilibrium isotherms of the CH_4_ or CO_2_ on samples were determined at 273 and 298 K, respectively, under saturated vapor pressure.

#### 2.5.2. Multi-Constituent Adsorption Breakthrough Method

Multi-constituent adsorption breakthrough curves were measured using a BSD-MAB analyzer from Beishide Instrument Technology (Beijing) Co., Ltd. (Beijing, China) The length of the breakthrough column was 68 mm with an internal diameter of 6 mm. The samples were activated and purged at 120 °C for 60 min using He with a purge gas flow of 30 sccm before measurement. Then, the feed gas containing CH_4_/CO_2_ (50/50, *v*/*v*) passed through the breakthrough column with a flow of 5 sccm at 273 K.

#### 2.5.3. Adsorption-Desorption Cycles Measurement Method

Adsorption–desorption cycle measurements were measured using a BSD-VVS and DVS gravimetric analyzer from Beishide Instrument Technology (Beijing) Co., Ltd. The adsorption measurements of the samples were measured at 298 K. The samples were treated to a vacuum (100 Pa) operation at 25 °C for 1 h during the desorption. The adsorption–desorption cycles were replicated up to 4 times under identical circumstances.

## 3. Results and Discussion

### 3.1. Structural Characterizations of the CPs and Their Crystallization Kinetics

The XRD patterns were used to investigate the influence of the content of additive OPD on the structures of CPs. As shown in Figure 1A, the XRD pattern of the parent CP (Figure 1(Aa)) exhibited well-defined peaks, indexed as (020), (200), (111), (13-1), (131), (22-2), (42-2), (151), (62-1) and (061), being consistent with those reported in the literature [41]. The intensity of the diffractive peak located at 2*θ* = 11.2° revealed the (200) reflection of HEU structures, which can be used to calculate the interlayer distances along a certain direction [44,45]. As can be seen in Figure 1(Ab,Ac), the intensity of the (200) peak of the synthesized CPs presented decreased trends with the increasing content of OPD from 1 to 3%; this variation tendency reflects the loss of long-range order in a certain direction. Figure 1(Ac,Ad) showed that the peak intensity (200) of CP-0.03 and CP-0.05 presented an unobvious change when the OPD additive content was around 3~5%. However, when the content of the additive OPD was more than 10%, the peak intensity (200) of CP-0.10 and CP-0.50 presented a decreased trend (shown in Figure 1(Ae,Af). Furthermore, the additional peaks (2θ = 12.5° and 28.2°) obviously appeared in CP-0.10 and CP-0.50, being ascribed to the impurity of phillipsite. Similar phenomena were also reported by Alvaro-Munoz et al. [46]. These observations indicated that the nucleation process in the synthesis of CP might be disrupted when the additive OPD content was more than 10%.

In order to provide an in-depth understanding of the effect of OPD in the synthesis of CP, we further explored its crystallization kinetics. According to the reported literature [47], the crystallization progress of traditional zeolites usually involves induction, growth and stable periods. The induction stage is a time when approximately 15% crystallinity passes through, belonging to the nucleation stage of CP crystals, while the growth period is described as the differences between the time taken to achieve steady crystallinity and the induction time (*t*_0_). Therefore, the relative value of the sum of the intensities of ten peaks in the XRD patterns was used to determine the crystallinity of the synthesized CPs: (020), (200), (111), (13-1), (131), (22-2), (42-2), (151), (62-1) and (061). The relative crystallinities of various synthesized CPs were determined in accordance with the normalized crystallinity of CP prepared at a crystallized time of 72 h at 150 °C without OPD.

Figure 2 shows the crystallization kinetics curves of the synthetic CPs under four different synthetic treatments (0, 3, 6 and 9% OPD added, respectively) at three temperatures (140, 150 and 170 °C).

As can be seen in Figure 2A, the *t*_0_ value gradually decreased with the increased contents of OPD at 140 °C, from 62 h without OPD (Figure 2(Aa)) to 34, 26 and 20 h after adding OPD of around 3% (Figure 2(Ab)), 6% (Figure 2(Ac)) and 9% (Figure 2(Ad)), respectively. Similarly, the *t*_0_ value decreased at 150 °C (Figure 2B) from 28 h for CP (Figure 2(Ba)) to 22 h for CP-0.03 (Figure 2(Bb)), 19 h for CP-0.06 (Figure 2(Bc)) and 17 h for CP-0.09 (Figure 2(Bd)), respectively. Obviously, the addition of OPD significantly shortens the induction periods of the synthesized CPs, which is beneficial to the promotion of the crystal nucleus. However, when the crystallized temperature reached 170 °C (Figure 2C), the slightly shortened *t*_0_ values for all the synthesized CPs corresponded to 18 (Figure 2(Ca)), 15 (Figure 2(Cb)), 17 (Figure 2(Cc)) and 14 h (Figure 2(Cd)), respectively. The possible reason is due to the rapid growth of crystal nuclei and the decline of crystallinity when the crystallization temperature improved to 170 °C.

To further elucidate the crystallization mechanism of the synthesized CPs in detail, the activation energy during the induction (*E_n_*) and growth (*E_g_*) periods in the synthesis of CPs was estimated on the basis of the Arrhenius equation.

The *E_n_* values and frequency factor (ln*A_n_*) were calculated using the nucleation rate (*1/t*_0_) and temperature based on Equation (2) [48]:(2)$$\ln\frac{1}{t_{0}}=\ln A_{n}-\frac{E_{n}}{RT}\;\;\;$$
where *R* is the ideal gas constant and *T* is the absolute temperature (K).

Equation (3) is used to calculate the activation energy of the growth period (*E_g_*) [49], and the slope at the crystallization curve’s steepest spot can be used to determine the rate constant (*k_max_*) as:(3)$$\ln k_{max}=\ln A_{g}-\frac{E_{g}}{RT}$$
where *A_g_* is the growth stage’s frequency factor.

Table 1 summarizes various parameters of the *E_n_*, *t*_0_ and *k_max_* values during the crystallization of the synthesized CPs. As can be seen, the *E_n_* values of the synthesized CPs with OPD were higher than the *E_g_* values, demonstrating that the hydrothermal crystallization process included a controlled stage called nucleation, which is the same as that of the CP synthesized without additive OPD. Similarly, the *E_n_* value of CP-0.03, CP-0.06 and CP-0.09 were 41.0, 21.0 and 18.0 kJ/mol, respectively, showing declined trends with the increase in the content of OPD, but lower than that of the CP obtained without additive OPD (61.5 kJ/mol). The *E_g_* values of the CP synthesized with and without additive OPD were almost the same during the growth periods, ranging from 16.0 to 19.5 kJ/mol. Therefore, we can conclude that the OPD plays a role in promoting the formation of the crystal nucleus due to accelerating the polymerization of silicate and aluminate species. In contrast, the effect of OPD on the growth procedure is awfully weak. Overall, the results stated above show that OPD addition can speed up crystallization during the induction period, which may be a key to modulating the particle sizes and morphologies of the synthesized CPs.

Figure 3 presents the morphologies of the synthesized CPs with different contents of OPD. As can be seen in Figure 3(Aa), the parent CP synthesized without OPD exhibited plate-like particles about 4 μm in size, while the images of the other samples in the presence of OPD, as seen in Figure 3(Ab–Ae), displayed different morphologies as compared with that of the parent CP (Figure 3(Aa)), where all others presented a flower-like shape. Correspondingly, the thickness of these samples obviously decreased, indicating that the additive OPD was conducive to the loss of long-range order in a certain direction. Meanwhile, the increase in the additive OPD content was conducive to reducing the lateral size of the flower-like CP from 4 μm for CP-0.01 synthesized with 1% of OPD (Figure 3(Ab)) to 2~3 μm for CP-0.03 and CP-0.05 and CP-0.1 (Figure 3(Ac–Ae)). The possible reason was that the crystallization process was accelerated with the increase in additive OPD content.

The fractal structural evolutions of the synthesized CPs with different contents of OPD were elucidated using the SAXS patterns. The following surface fractal dimension (*D_s_*) values could be calculated using the ln[*I*(*q*)] against ln(*q*) scattering profiles, as shown in Figure 4A, where the straight lines in the *q* areas were fitted using the linear least-squares approach. The *D_s_* value gradually increased with the increasing contents of OPD, from 2.02 for CP without OPD to 2.09 for CP-0.01, 2.12 for CP-0.02 and 2.31 for CP-0.03, indicating that the surfaces of the synthesized CPs in the presence of OPD became rougher with the increasing content of OPD. Then, the *D_s_* value (2.34 for CP-0.04 and 2.37 for CP-0.05) was no longer an obvious change with the increasing continuous contents of OPD, implying that the disorder of the synthesized CPs may reach an upper limit.

Further details on the geometry and shape characteristics of the synthesized CPs are revealed with the pair distance distribution function (PDDF) curves that are obtained from the SAXS patterns. As shown in Figure 4B, the PDDF curves for all the samples were short of perfect symmetry, implying that the particles of the synthesized CPs may have a flake-like morphology [50], being consistent with the results of the SEM images (Figure 3(Aa–Ae)). The maximum value of the intersection of the PDDF curve with the *X*-axis was 99 nm for the CP without OPD (Figure 4(Ba)) but decreased to 89 nm for CP-0.01 (Figure 4(Bb)), 83 nm for CP-0.02 (Figure 4(Bc)) and 75 nm for CP-0.03 (Figure 4(Bd)); however, that of CP-0.04 and CP-0.05 (Figure 4(Be,Bf)) was almost similar to that of CP-0.03. These results may indicate that the thickness of the random lamellae of the synthesized CPs was thinner with the increasing contents of OPD than that of the CP without OPD, but it did not become thinner until a particular content (more than 3%). These phenomena are consistent with the demonstrations of the SEM pictures (Figure 3) and XRD patterns (Figure 1).

The crystallization process of the synthesized CPs with additive OPD was also investigated using the XRD patterns, FT-IR spectra, SEM images, ^29^Si-NMR profiles and SAXS patterns. The diffuse peaks at 2 *θ* of 10–40° were visible in the XRD patterns of the CPs synthesized with a crystallization period of 12 h, as shown in Figure 1(Ba), suggesting the presence of the amorphous phases [51]. The characteristic peaks appeared at 9.9, 11.2 and 22.3°, indexed as (020), (200) and (131), when the prolonged crystallization time was up to 36 h (Figure 1(Bb)), indicating the formations of the HEU structures of the CPs.

Additionally, as shown in Appendix A of their FT-IR spectra in the Electronic Supporting Information (ESI) Section, the absorption band at 1638 cm^−1^ was connected to the deformation and vibration of the H_2_O molecule [52]. The peaks that appeared at 1205 cm^−1^ and 1062 cm^−1^ were ascribed to the asymmetric internal T–O–T stretching vibrations of the tetrahedral atoms in CP [53]. The absorption peak located at 605 cm^−1^ was related to the internal bending vibration of the T–O bond. The absorption peak approximately at 465 was ascribed to the T–O bond’s internal bending vibration in the TO_4_ structure. Particularly, the absorption bands centered at 605 and 1215 cm^−1^ were significantly strengthened with the increase in crystallization time from 12 to 72 h. These phenomena are consistent with the demonstrations of the XRD patterns (Figure 1(Ba,Bb)). Correspondingly, Figure 3(Ba) shows the amorphous particles with a crystallization time of 12 h. When the crystallization time was further extended to 36 h, the amorphous particles interconnected into lamellar structures (Figure 3(Bb)). The flower-like particles in size of more than 3 μm were observed at the crystallization time of 72 h (Figure 3(Ac)).

The fractal evolutions and the PDDF curves of the synthesized CPs with different crystallization times were investigated using the SAXS patterns. As can be seen in Figure 4C, as the crystallization time increased from 0 to 24 h, the *Ds* values of the synthesized CPs decreased from 2.31 to 2.03, implying the sol-gel particles of the aluminosilicates species became gradually smooth during the early crystallization procedures. However, the *Ds* values increased to 2.28 for 36 h and 2.36 for 48 h, suggesting that the particles became rougher since the amorphous particles transformed into flower-like structures on the basis of the SEM images (Figure 3(Bb)). After that, the *Ds* values were kept nearly constant with a further crystallization time of 48–72 h.

As shown in Figure 4D, the PDDF curves of the synthesized CPs became more asymmetrical with the increase in the crystallization time than that of 0 h (Figure 4(Da)) with an approximate symmetric shape, indicating the generation of the irregular morphologies with the crystallization times. Meanwhile, the maximum value of the intersections of the PDDF curves (Figure 4(Da–Dc)) with the *x*-axis was almost 110 nm for CPs synthesized with a crystallization time of 0–24 h, probably implying that the sizes of the amorphous particles were within this range, being consistent with the demonstrations of SEM images (Figure 3(Ba)). However, the maximum (Figure 4(Dd–Dg)) values declined to almost 75 nm with the prolonged crystallization time of 36–72 h, indicating the lamella thickness of these synthesized CPs was around 75 nm [54]. Notably, it cannot represent the size of the whole particles because the SAXS method can only detect small particles less than 100 nm [55].

Figure 5 illustrates the ^29^Si-NMR profiles of CP-0.03 obtained at different crystallization times. Since the CP with the HEU frameworks belongs to a monoclinic zeolite with five distinct T sites, a major overlapping should occur in the ^29^Si-NMR patterns due to different T–T distances and T–O–T angles [56]. As can be seen in Figure 5A, five resonance peaks can be identified after deconvolution in the early stages of the crystallization of 12 h, whereas the broad resonances centered at −85, −95, −100, −105 and −110 ppm were characteristic of Si(4Al), Si(3Al), Si(2Al), Si(1Al) and Si(0Al) silicon environments, which may be due to the rapid combination of aluminate and silicate [51]. The morphology of the crystallization of 12 h showed the amorphous particles of the aluminosilicates sol-gel in the SEM image (Figure 3(Ba)). With the extension of the crystallization time up to 36 h (Figure 5B), the disappearances of the Si(4Al) and Si(3Al) suggested that more silicates were embedded in the aluminosilicate networks and then emerged the appearances of Si(1Al) mostly in the subsequent crystallization of 36 h, being consistent with the reported literature [57,58]. Correspondingly, the amorphous particles interconnected into lamellar structures on the basis of the SEM image (Figure 3(Bb)). After that, three resonance lines mainly involved Si(2Al), Si(1Al) and Si(0Al) silicon surroundings as the crystallization time extended to 72 h (Figure 5C). Meanwhile, the flower-like structures were observed in the SEM images (Figure 3(Ac)). On the basis of Al-Yassir’s report [59] and Equation (1), the calculated Si/Al ratios of all samples were 2.98 (12 h), 4.42 (36 h) and 4.33 (72 h), respectively, showing increased tendencies with the crystallization time of before 36 h, but the stable periods subsequently.

### 3.2. Structure and Morphology Characterizations of the Synthesized CPs

The XRD patterns of the synthesized CPs before and after modification and ultrasonic/washing can be used to qualitatively assess their differences in the lamellar structures. As shown in Figure 1(Ac,Bc,Bd), the XRD patterns of all samples exhibited well-defined peaks at (020), (200), (111), (13-1), (131), (22-2), (42-2), (151), (62-1) and (061), in good agreement with that of the parent CP (Figure 1(Aa)), indicating that the CP can be retained intact in the HEU structures during the post treatments. Meanwhile, the intensity of the (200) diffractive peak can be used to reflect the interlayer distances along a certain direction, and, obviously, their peak intensity of (200) decreased significantly after the ultrasonic treatment of CP-0.03-U (Figure 1(Bc)) and the subsequent washing of CP-0.03-U-E (Figure 1(Bd)), but that of the CP-0.03 (Figure 1(Ac)) decreased slightly. These observations suggest that the effects of the post-treatments, particularly, with ultrasonic and washing, on the long-range order of the CPs in a certain direction are obvious, which is conducive to the formations of the random lamellae. The random lamellae proposed in our work mainly refer to the macroscopic morphology of the synthetic CP rather than its HEU microscopic structure. Similar results were also demonstrated by Schmidt et al. [60], who reported that the topotactic condensation of a layered aluminosilicate material (CIT-8P), used as a precursor for the preparation of a high-silica HEU (CIT-8), resulted in the CIT-8P and CIT-8 exhibiting a plate-like morphology, but their plate thickness was unusual. These observations were also verified in other literature [61]. Therefore, we can conclude that the lamellae of the HEU-grouped CP can be modulated in a certain direction.

Using OPD as a modifier, the random lamellae of the synthetic CP were prepared with post-ultrasonic treatment and alcohol washing. Compared with the preparation process reported in the literature, no high temperature and acidification conditions were required, which was conducive to avoiding the destruction of the HEU skeletons.

Figure 3 shows the SEM/TEM images of the synthesized CPs to further investigate the effect of the ultrasonic treatment and alcohol-washing process on their particle sizes and related morphologies. As can be seen in Figure 3(Af), CP-0.03 showed a thickness of each nanoflake of around 15–20 n; however, the irregular random lamellae with a smaller size of around 0.5–1 μm for CP-0.03-U (Figure 3(Bc)) and 0.2–0.5 μm for CP-0.03-U-E (Figure 3(Bd)) were observed after the ultrasonic and washing treatments. The probable reason is that the force generated by the collapse of bubbles (originating from the used solvent in the presence of the ultrasonic waves) causes random lamellae to be formed [62,63,64]. As shown in Figure 3(Be,Bf), the TEM images of CP-0.03-U presented the multi-layer irregular random lamellae, and CP-0.03-U-E obtained by further reducing the number of lamellae eventually showed the thinner lamellar structures.

As reported by Baerlocher and Mccuske [65], the HEU frameworks consist of 10-membered rings (0.31 × 0.75 nm) in the (001) direction and 8-membered rings (0.36 × 0.46 nm) in the (001) direction. Meanwhile, another set of 8-membered rings along the (100) direction with dimensions of 0.28 × 0.74 nm also existed, as mentioned in the Introduction Section. On the basis of the diffractive (200) peak profile (as shown in Figure 1), and the Bragg Equation (4), the interplanar distance of the (200) crystal plane was calculated to be around 1.5828 nm using:(4)2dsinθ=nλ
where *d* is the spacing of crystal planes, *θ* is the angle between the incident X-ray and the corresponding crystal plane, *n* is the diffraction series, and *λ* is the wavelength (0.154056 nm) of X-rays.

In the present work, the thickness of the nanosheets composed of the lamellae could be determined with TEM observation, but our reported nanosheets for the HEU structures were not accurate enough to discuss their results. One of the main reasons is that it is very difficult for us to capture the cross-sections of the lamellae using the TEM experiments presently. Therefore, we speculated that the nanosheet thickness of the lamellae should be less than 1.5828 nm along the (200) direction.

Figure 4E illustrates the fractal structure evolution of the various CPs and the random lamellae of the synthetic CP. As can be seen, the *D_s_* values determined from the ln[*I*(*q*)] versus ln(*q*) scattering profile in the *q* regions presented the fractal features of all CPs. The *Ds* value increased from 2.02 for the parent CP (Figure 4(Ea)) to 2.31 for CP-0.03 (Figure 4(Eb)), 2.46 for CP-0.03-U (Figure 4(Ec)) and 2.55 for CP-0.03-U-E (Figure 4(Ed)), indicating that the synthesized CPs became more irregular and rough during the ultrasonic and washing treatments. As can be seen in Figure 4F, the asymmetrical shapes of the PDDF profiles suggest that these synthesized CPs represented a disk-like morphology along with OPD modification and further ultrasonication/washing. Furthermore, the maximum value of the intersection of the PDDF curve with the *x*-axis was 99 nm for the parent CP (Figure 4(Fa)) but decreased to 76 nm for CP-0.03 (Figure 4(Fb)), 59 nm for CP-0.03-U (Figure 4(Fc)) and 51 nm for CP-0.03-U-E (Figure 4(Fd)). These results again implied that the nanoflake thickness of CP-0.03 became gradually thinner with ultrasonic and washing than that of parent CP, in good agreement with the demonstrations of the SEM and TEM images.

Figure 5D illustrates the ^29^Si-NMR profiles of CP-0.03-U-E. As can be seen, the resonance signals around −85, −95, −100, −105 and −110 ppm should be assigned to the Si(4Al), Si(3Al), Si(2Al), Si(1Al) and Si(0Al) silicon environments, respectively. Meanwhile, its *n*(Si/Al) value of 4.34 presented no obvious change as compared with that of CP-0.03 with a crystallization time of 72 h (Figure 5C). These results further demonstrate that the structural integrity of the HEU frameworks could be preserved during the random lamellae formation process, showing significant advantages as compared with some literature reports [33,35], such as avoiding the occurrences of the desilication phenomena and acidification.

Appendix A presents the TG curves of the various CPs synthesized before and after OPD modification and ultrasonic/washing. As can be seen, the weight loss profiles of the synthesized CP could be divided into two stages: the first one occurred when the temperature was up to 300 °C, belonging to the removal of physically adsorbed water, and the second happened during the temperature range from 300 to 800 °C, corresponding to the desorption of more strongly associated chemisorbed water [66]. Although the weight loss of CP-0.03 (Appendix A) at 300–800 °C was approximately 0.5%, almost the same as that of the parent CP (Appendix A), its weight loss at 30–300 °C was around 8.2%, which was higher than that of the parent CP; the main reason was due to the decompositions of residual OPD and the removal of physically adsorbed water. However, the weight loss of CP-0.03-U (Appendix A) and CP-0.03-U-E (Appendix A) at 30–300 °C was less than that of CP-0.03, due to the removal of a little residual OPD and dehydroxylation. It seems reasonable to assume that the removal of OPD is conducive to the appearance of unordered lamellar structures, which can be evidenced by the small decrease in peak (200) intensity in the XRD patterns (Figure 1).

The N_2_adsorption–desorption isotherms and pore size distribution of the synthesized CPs are shown in Appendix A, and their textural parameters were collected in Appendix A. As can be seen in Appendix A, all of the isotherms had the typical type-I curves with an H3-type hysteresis loop at 0.80 < *P/P*_0_ < 0.98, due to the presence of the mesopores structures originating from the accumulation of the random lamellae of the synthetic CPs.

Specifically, the CP-0.03-U-E (Appendix A) had a higher specific surface area (36.7 m^2^·g^−1^) and larger mesoporous volume (0.115 mL·g^−1^), as compared with that of the parent CP without delamination (Appendix A).

However, the textural properties of the various synthetic CPs were poor (as shown in Appendix A), and the probable reasons were due to their random lamellas resulting in lower surface areas and pore volumes. As demonstrated in numerous reports [51,67,68,69], the BET surface area of the conventional clinoptilolite was around 20.0–71.6 m^2^·g^−1^, mainly depending on the synthetic methods and the various morphology.

### 3.3. CO_2_ and CH_4_ Adsorption Performances

Figure 6 presents the CO_2_ and CH_4_ adsorption isotherms of the various CPs at 273 and 298 K. As shown in Figure 6A,B, the adsorbed capacity of CO_2_ for all CPs was significantly greater than that of CH_4_. On the on hand, the CO_2_ adsorbed capacity of CP-0.03-U-E was up to around 2.14 mmol·g^−1^ at 273 K and 1.84 mmol·g^−1^ at 298 K, which was higher than that of other CPs (parent CP, CP-0.03 and CP-0.03-U). Although the CO_2_ adsorbed capacity at 273 K decreased in the following order: CP-0.03-U-E > CP-0.03-U > CP-0.03 > parent CP (Figure 6A), the adsorption capacities of the other CPs at 298 K were almost the same (Figure 6B). On the other hand, the CH_4_ adsorption isotherms at 273 and 298 K, presented in Figure 6C,D, show that the CH_4_ adsorption capacities of all CPs were around 0.65–0.7 mmol/g at 273 K (Figure 6C) and 0.45–0.52 mmol/g at 298 K (Figure 6D), showing no significant differences. The separation factors at various temperatures were determined using the Freundlich-Langmuir (F-L) isotherm [70] to further elucidate the influence of random lamellae on the performance of CO_2_/CH_4_ separation:(5)$$Q=\frac{q_{s}{KC}^{n}}{{1+KC}^{n}}\;$$
where *Q* is the moles adsorbed content (mmol/g), *q_s_* is the saturation adsorption capacity in the moles (mmol/g), *K* and *n* are Freundlich–Langmuir constants, and *C* is the relative pressure.

The selectivity factors of CO_2_/CH_4_ are illustrated in Figure 6E,F. As can be seen in Figure 6E, the selectivity factor of CO_2_/CH_4_ of CP-0.03-U-E at 273 K was up to 3.6 at a pressure of 1.0 bar, which was much higher than that of the parent CP, CP-0.03 and CP-0.03-U. Similarly, Figure 6F presents that the separation factor of CP-0.03-U-E (3.4) was higher than that of the other CPs at a pressure of 1.0 bar and 298 K. Additionally, the separation factor of CP-0.03-U-E calculated at 273 K was slightly higher than that at 298 K, being consistent with the findings of Pour & Sharifnia [39].

The adsorption heats of the various CPs were determined based on the Clausius–Clapeyron equation [71,72]:(6)$$\ln\frac{P_{1}}{P_{2}}=\frac{\Delta H_{vap}}{R}(\frac{1}{T_{1}}-\frac{1}{T_{2}})\;$$
where *P*_1_ and *P*_2_ are the relative pressures at *T*_1_ and *T*_2_, respectively, Δ*H_VAP_* is the isochoric adsorption heat of CO_2_ or CH_4_, and *R* is the gas constant (8.314 J·mol^−1^·K^−1^).

The adsorption heats of CO_2_ and CH_4_ for the various CPs are presented in Figure 7. Obviously, their adsorption of CO_2_ and CH_4_ was an exothermic process. As shown in Figure 7A, the CO_2_ adsorption heats of CP-0.03, CP-0.03-U and CP-0.03-U-E were more than that of the parent CP. The main reasons are related to their morphology and the CO_2_ adsorption sites. As seen in Appendix A, the specific surface area gradually increased from 20.4 m^2^·g^−1^ for the parent CP to 27.1 m^2^·g^−1^ for CP-0.03, 32.8 m^2^·g^−1^ for CP-0.03-U and 36.7 m^2^·g^−1^ for CP-0.03-U-E; correspondingly, the morphology was from plate-like particles for the parent CP (Figure 3(Aa)) to flower-like particles for CP-0.03 (Figure 3(Ac)), irregular lamellae for CP-0.03-U (Figure 3(Bc)) and random lamellae for CP-0.03-U-E (Figure 3(Bd)). Obviously, CP-0.03, CP-0.03-U and CP-0.03-U-E with larger specific surface areas might easily expose more CO_2_ adsorption sites, resulting in that the CO_2_ adsorption performance of CP-0.03, CP-0.03-U and CP-0.03-U-E was stronger than that of parent CP. Therefore, the CO_2_ adsorption heats of CP-0.03, CP-0.03-U and CP-0.03-U-E (Figure 7(Ab–Ad)) were higher than that of the parent CP (Figure 7(Aa)). Particularly, the random lamellae of CP-0.03-U-E with a larger specific surface area presented a higher CO_2_ adsorption heat (30.5–35.7 mmol/g) than that of the parent CP (0–22.9 mmol/g). Similar demonstrations were also reported by Jia et al. [73]. However, the adsorption heats of CP-0.03 and CP-0.03-U were more than that of CP-0.03-U-E, probably because of the presence of the residual OPD in CP-0.03 and CP-0.03-U [74,75]. Figure 7B showed that all CPs had nearly the same adsorption heats of CH_4_ but less than that of CO_2_.

Figure 8 illustrates the breakthrough curves of CO_2_ and CH_4_ on the various CPs at 273 K with a feed gas flow rate of 5 sccm, whereas the feed gas consists of 50% CO_2_ and 50% CH_4_. Breakthrough time is classified as the time required for 5% of the CO_2_ input fraction to appear in the output stream [76]. Obviously, the breakthrough time of the various CPs was longer for CO_2_ but shorter for CH_4_. In this test, the effluent stream from the column only contained CH_4_ before reaching the breakthrough time of CO_2_, indicating that CO_2_ was more strongly adsorbed than CH_4_ on the CPs, in other words, the CO_2_ was preferentially adsorbed by the CPs. As can be seen in Figure 8D, the breakthrough time of CO_2_ on CP-0.03-U-E was about 1055 s, which was higher than that on the other CPs, such as 758 s for the parent CP, 959 s for CP-0.03 and 976 s for CP-0.03-U. Correspondingly, the dynamic adsorption capacities of CO_2_ and CH_4_ on CP-0.03-U-E were 2.20 and 1.19 mmol/g, which were higher than that on the parent CP (1.67 and 0.97 mmol/g), CP-0.03 (2.02 and 1.14 mmol/g) and CP-0.03-U (2.05 and 1.15 mmol/g). Obviously, these results demonstrate that CP-0.03-U-E had a higher adsorption capacity of CO_2_ than the other CPs with the postponement of the CO_2_-saturated column.

Figure 9 shows the stability and reversibility of the CO_2_ adsorption performances with the various CPs via CO_2_ adsorption–desorption cycles. It is observed that the CO_2_ adsorbed amount for all CPs is nearly constant after the fourth cycle, indicating excellent stability and reversibility, which enable effective CP regeneration and recycling.

## 4. Conclusions

A random lamella of the synthetic CP was successfully prepared with the aid of OPD as a modifier using the conventional hydrothermal method. Its crystallization kinetics showed that the addition of OPD was beneficial to the promotion of the nucleation process, which was a controlled step during the induced stage and subsequent growth periods. Meanwhile, the random lamellae were obviously obtained with further ultrasonic and washing treatments. Specifically, the SAXS patterns strongly demonstrated the surface fractal evolution of the aluminosilicate sol-gel from smooth spheres to irregular roughness in the hydrothermal crystallization duration. The CO_2_ and CH_4_ adsorption performances of the various CPs at 273 and 298 K presented that the CO_2_ uptakes were much higher than the CH_4_ adsorbed capacities; particularly, the CO_2_ adsorption capacity of the random lamellae of the synthetic CP at 273K was around 2.14 mmol/g, which was higher than that of the parent CP (1.48 mmol/g). The selectivity factor of CO_2_/CH_4_ increased from 2.4 for the parent CP to 3.4 for the random lamellae of the synthetic CP. The preliminary breakthrough explorations showed a longer time for CO_2_ than that for CH_4_, in addition to the excellent cycling behaviors and the remarkable repeatability of adsorption performance. These demonstrations elucidated that an amphipathic solvent-assisted synthetic strategy can regulate the random lamellae of the synthetic CP for the adsorption and separation of CO_2_ and CH_4_.

## Figures and Tables

**Figure 1 nanomaterials-13-01942-f001:**
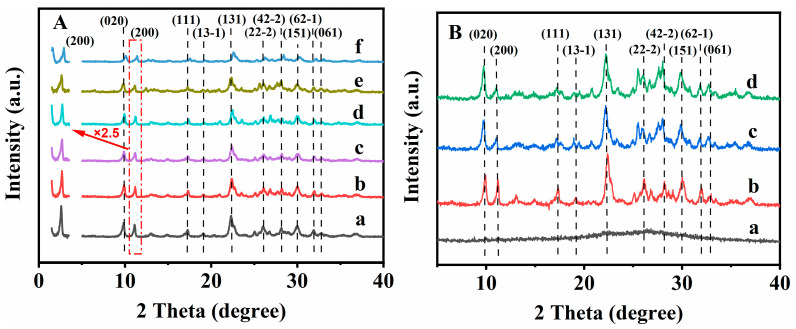
XRD patterns of (**A**): parent CP (a), CP-0.01 (b), CP-0.03 (c), CP-0.05 (d), CP-0.10 (e) and CP-0.50 (f); (**B**): CP-0.03 synthesized CPs with different crystallization times: 12 h (a), 36 h (b), CP-0.03-U (c) and CP-0.03-U-E (d).

**Figure 2 nanomaterials-13-01942-f002:**
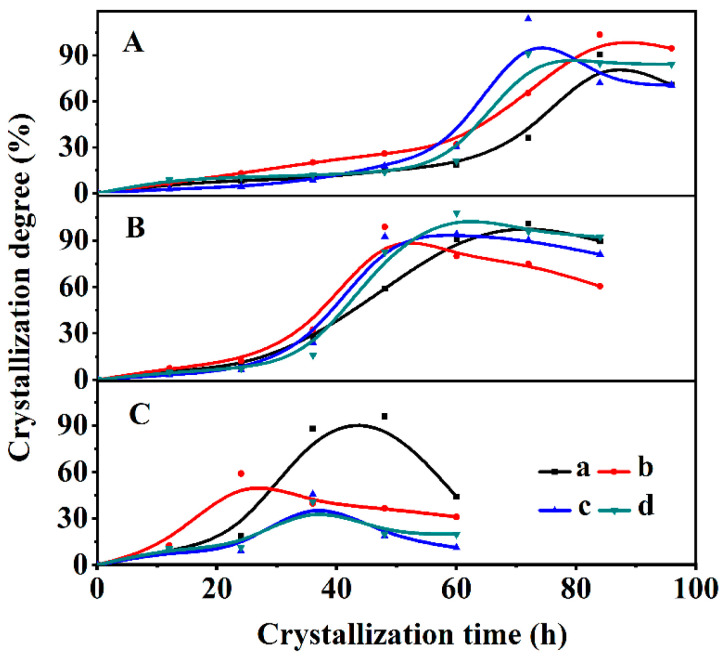
Crystallization kinetic curves of the synthesized CPs with additive OPD at 140 °C (**A**), 150 °C (**B**) and 170 °C (**C**) for parent CP (a), CP-0.03 (b), CP-0.06 (c) and CP-0.09 (d).

**Figure 3 nanomaterials-13-01942-f003:**
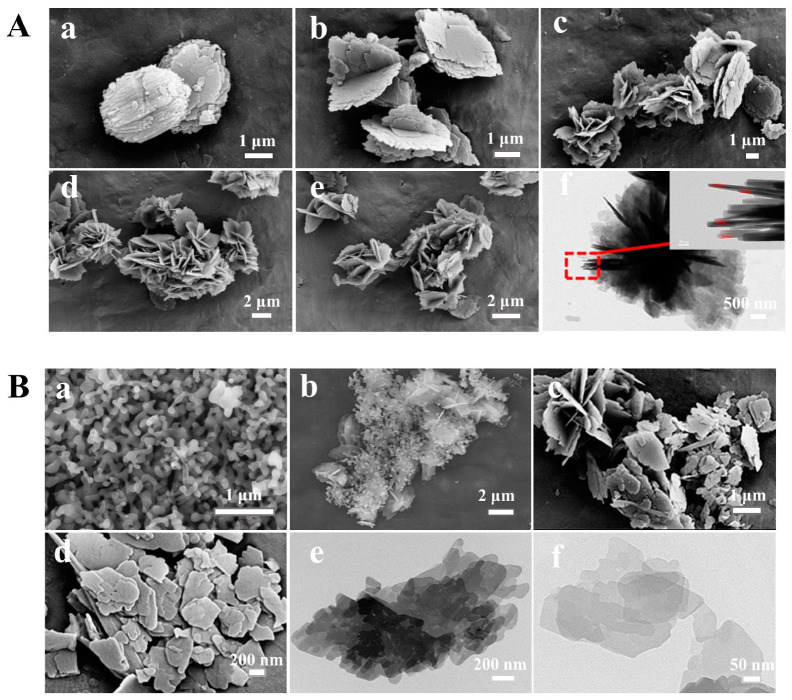
SEM images (**A**) of the parent CP (**a**), CP-0.01 (**b**), CP-0.03 (**c**), CP-0.05 (**d**) and CP-0.1 (**e**) and TEM images of CP-0.03 (**f**); SEM images (**B**) of CP-0.03 with different crystallization times: 12 h (**a**) and 36 h (**b**), and CP-0.03-U (**c**) and CP-0.03-U-E (**d**); and TEM images (**B**) of CP-0.03-U (**e**) and CP-0.03-U-E (**f**).

**Figure 4 nanomaterials-13-01942-f004:**
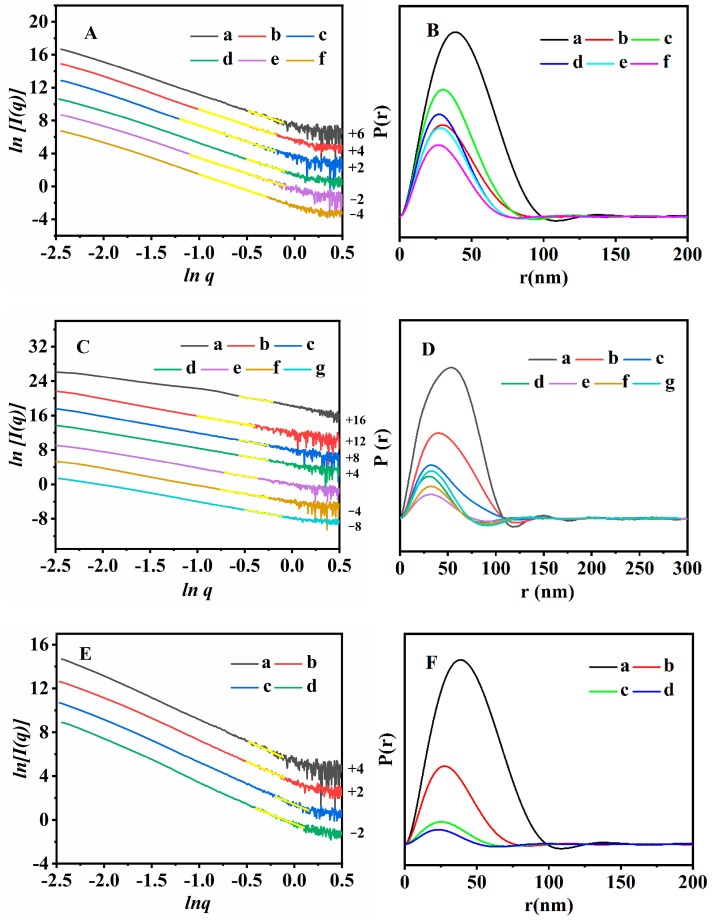
Shifted scattering curves (**A**,**C**,**E**) and the PDDF profiles (**B**,**D**,**F**) of (**A**,**B**): Parent CP (a), CP-0.01 (b), CP-0.02 (c), CP-0.03 (d), CP-0.04 (e) and CP-0.05 (f); (**C**,**D**): CP-0.03 with various crystallization times: 0 h (a), 12 h (b), 24 h (c), 36 h (d), 48 h (e), 60 h (f) and 72 h (g); and (**E**,**F**): parent CP (a), CP-0.03 (b), CP-0.03-U (c) and CP-0.03-U-E (d).

**Figure 5 nanomaterials-13-01942-f005:**
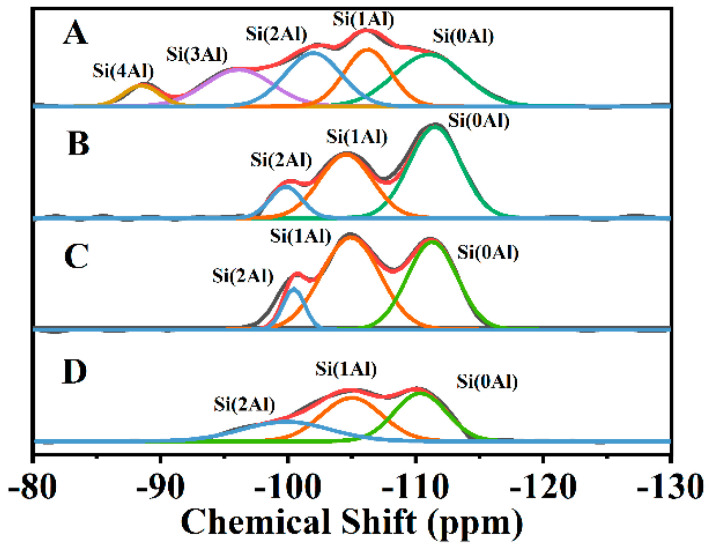
The ^29^Si-NMR patterns of the CP-0.03 synthesized CPs with different crystallization times: 12 h (**A**), 36 h (**B**) and 72 h (**C**) and CP-0.03-U-E (**D**).

**Figure 6 nanomaterials-13-01942-f006:**
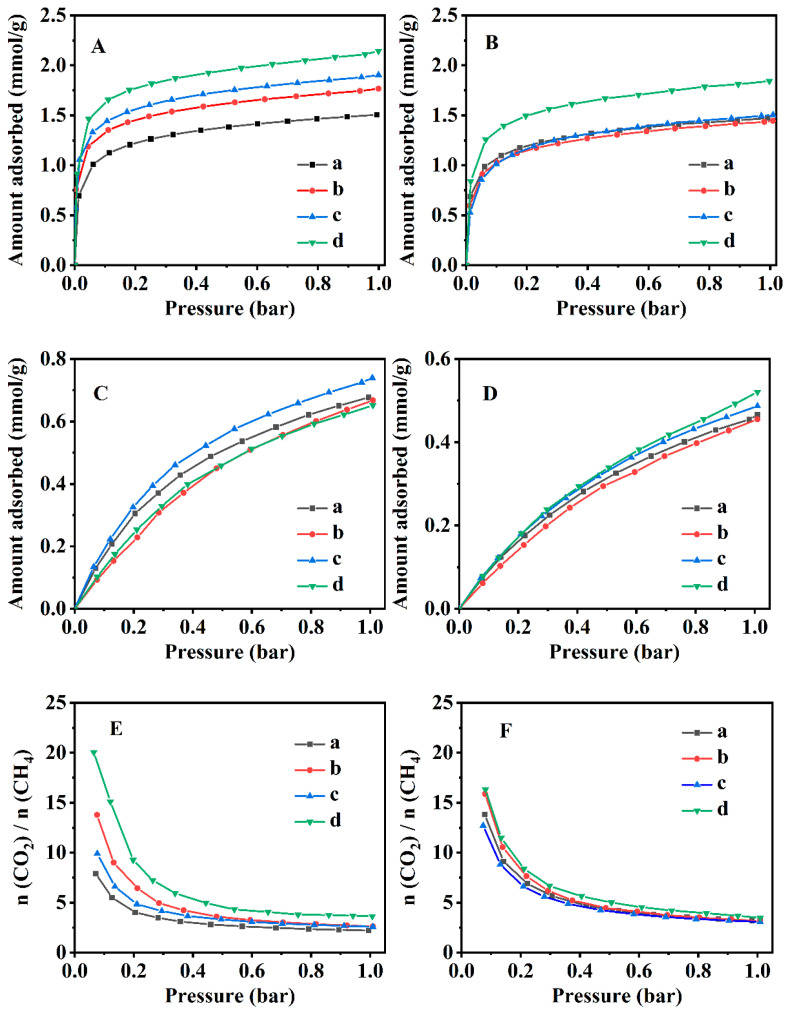
Equilibrium adsorbed isotherms of CO_2_ and CH_4_ using various CPs as the adsorbents at 273 K (**A**,**C**) and 298 K (**B**,**D**), respectively. CO_2_/CH_4_ selectivity of the CPs at 273 K (**E**) and 298 K (**F**). Parent CP (a), CP-0.03 (b), CP-0.03-U (c) and CP-0.03-U-E (d).

**Figure 7 nanomaterials-13-01942-f007:**
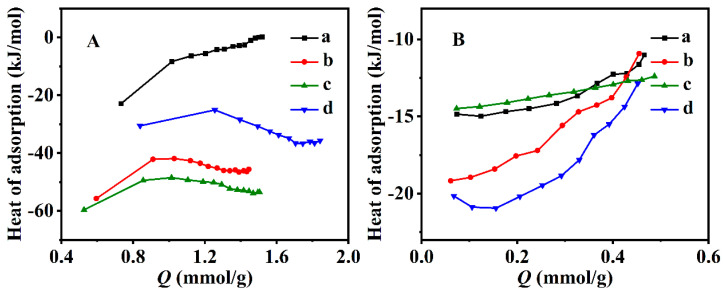
Adsorption heats of CO_2_ (**A**) and CH_4_ (**B**) on the CPs: parent CP (a), CP-0.03 (b), CP-0.03-U (c) and CP-0.03-U-E (d).

**Figure 8 nanomaterials-13-01942-f008:**
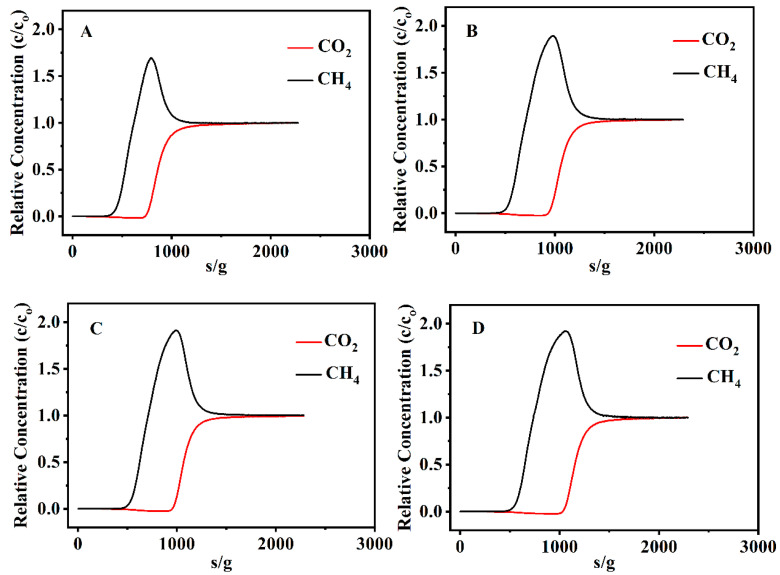
Breakthrough curves of CO_2_ and CH_4_ on the various CPs at 273 K, (**A**) parent CP, (**B**) CP-0.03, (**C**) CP-0.03-U and (**D**) CP-0.03-U-E.

**Figure 9 nanomaterials-13-01942-f009:**
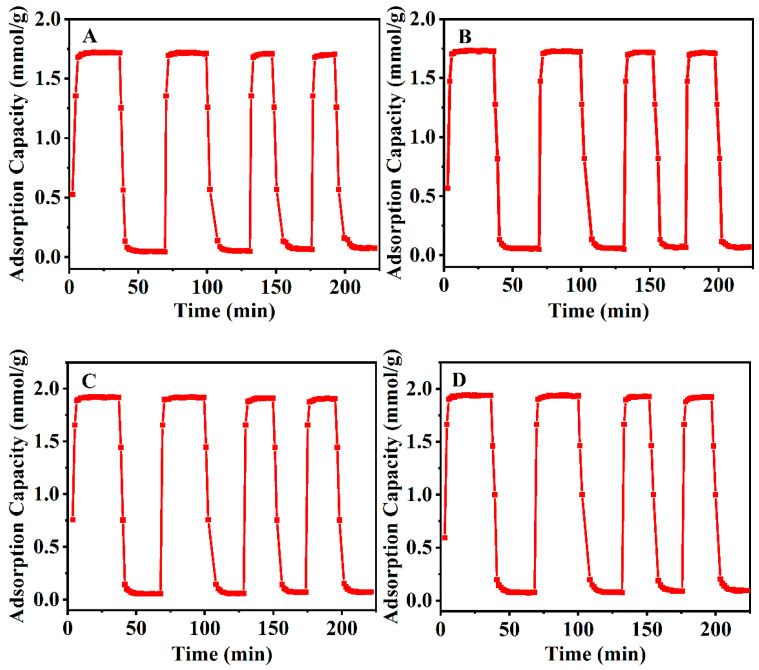
CO_2_ adsorption–desorption cycles of the CPs showing the stability and reversibility: (**A**) parent CP, (**B**) CP-0.03, (**C**) CP-0.03-U and (**D**) CP-0.03-U-E.

**Table 1 nanomaterials-13-01942-t001:** Summaries of various kinetic parameters in the crystallization procedures of the synthesized CPs.

Sample	T (°C)	Induction Period			Growth Period		
		*En* (kJ/mol)	ln*A_n_*	*t*_0_ (h)	*k_max_*	*E_g_* (kJ/mol)	ln*A_g_*
CP	140	61.5	13.7	62	1.7	17.5	5.5
	150			28	1.9		
	170			18	2.4		
CP-1,2-DMB-0.03	140	41.0	8.3	34	1.5	19.5	6.0
	150			22	1.7		
	170			15	2.2		
CP-1,2-DMB-0.06	140	21.0	2.8	26	2.7	17.5	6.1
	150			19	3.1		
	170			17	3.8		
CP-1,2-DMB-0.09	140	18.0	2.2	20	3.1	16.0	5.8
	150			17	3.6		
	170			14	4.3		

## Data Availability

Not applicable.

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
