# Peer review of "Amphipathic Solvent-Assisted Synthetic Strategy for Random Lamellae of the Clinoptilolites with Flower-like Morphology and Thinner Nanosheet for Adsorption and Separation of CO2 and CH4"

_nanomaterials, 2023, doi:10.3390/nano13131942_

Round 1
Reviewer 1 Report
General comments
There are many long sentences (lines 11-15, lines 30-33, lines 46-50, lines 68-72, lines 111-114, lines 223-226, lines 293-297 and so on) which make the text very tedious. Moreover, there are many phrasal, syntactical and grammar mistakes throughout the manuscript (such as in line 36, line 161, line 172, line 257, line 278 and many other). For example, there are many unnecessary commas (e.g. line 41 and 274 after while, line 44 etc. All commas before “and” must be removed). On the contrary there are more than one “and” in some sentences instead of using a comma (e.g. line 43). A lot of sentences must be rephrased in order to be more bramble (such as lines 128-130, lines 149-151, lines 274-276 and many other).
If there is any novelty of the work is not clearly stated.
Compare your best catalytic results with the literature.
Figure 1. It is not clear that the intensity of (200) peak stays steady by increasing ODP more than 3% as you claim in lines 220-223. On the contrary there is an obvious difference concerning 50% ODP content and that of 3%. If this is not the fact, please magnify this part of XRD diagrams (2 theta region between 10-12 degrees) to be clearer.
Figure 3. The comments for Fig3k-l are at the part regarding crystallization time effect, so in a very different page than the pictures are. It would be better to split Fig 3 to two figures with the two effects (ODP content and crystallization time). Same goes with XRD patterns.
Figure S2B. You claim the materials have mesoporosity. The shift at more than P/Po=0.85 is usually assigned to textural porosity. You should also add BJH pore size distribution graphs to verify the statement.
It would be better to add your best catalyst and its relative catalytic performance in both abstract and conclusions.
Author Response
Response to Reviewer 1

Reviewer 2 Report
In the presented article, hierarchical clinoptilolites were hydrothermally synthesized with the aid of o-Phenylenediamine. These materials are shown to be good for CH4/CO2 separation, which is an important industrial challenge. Besides many materials, shown to have such activity, such as zeolites, titanosilicates, carbon molecular sieves, MOFs etc., the hierarchical clinoptilolites (reported in the present article) with o-Phenylenediamine show gas adsorption and separation performances much higher than that of the clinoptilolites synthesized without added OPD. It has been shown that the crystallization kinetics is governed by the addition of OPD, which was a controlled step during the induced stage and subsequent growth periods. The presentation of the results is good and the conclusions are motivated by the results. The figures and tables are clear and understandable. Besides the good representation and analysis of the results, there are still some open questions, which was not clarified in the manuscript.
1) What is the reproducibility of the results, concerning the synthesis conditions? Did the authors repeat the synthesis in the same experimental conditions to prove it?
2) What is the reason for such a high CH4 adsorption in sample CP-0.03-U-E , shown in fig 6B at 273K vs. the other samples where the adsorption curves almost overlap? Why does this effect vanish at 298K (Figure 6D)?
3)The authors show that the adsorption heat of CO2 in samples b,c, and d, is higher than that of the parent material, but did not explain why? What is the reason for such stronger CO2 interaction?
Author Response
Response to Reviewer 2

Reviewer 3 Report
In the abstract the authors claim preparation of ‘hierarchical clinoptilolites.. with the aid of phenylenediamine … acting as a swelling agent… (that) was sandwiched between the lamellas of the precursor, enlarging their interlayer spacing and resulting in the loss of long-range order in the a direction.’ The term (layered zeolite) precursor usually denotes a 2D material with layer thickness about 1-3 nm, which can condense to form the 3D zeolite framework. The Introduction discusses at length the swelling and delamination of such precursors so it must be in this sense that the definition is applied herein. By extension, the idea of OPD being sandwiched between lamellae and ‘enlarging interlayer spacing’ makes only sense for such thin layers, not tens of nanometer thick (as in this work). The problem is that the obtained materials are not such precursors but simply thin crystal HEU frameworks with 3D connectivity. This is evident from the XRDs in Figure 1. The authors make a cardinal mistake by stating and using (line 405) that ‘…the intensity of (200) diffractive peak can be used to calculate the interlayer distances along a direction’, which is false because it is THE POSITION of this peak that reflects the interlayer distance. As the XRD peak positions, especially 200, do not change the reported products are not precursors but simply thin HEU. For comparison the authors should consult the report on zeolite CIT-8, which is a precursor with HEU layers. How the authors can claim swelling (line 400 and delamination (line 405) is simply inexplicable. The images in Figure 3, especially (j) should be analyzed quantatively with regard to layer thickness, which now is left to a reader to examine on his own. Images (k) and (l) are not informative as they present crystals face-on.
Given that the principal tenet of this work is false there is no point in deeper analysis of this manuscript, which must be revised first and adjusted to the changed perspective. The manuscript, especially abstract contains a number of catch words (hierarchical, precursor, delaminated, exfoliation) which may not be apply at all here and must be well justified (and defined).
Author Response
Response to Reviewer 3

Round 2
Reviewer 1 Report
It can be published
Author Response
Response to Reviewer 2:
Thank you very much.
Reviewer 3 Report
When talking about lamellae it is essential to provide upfront (and in the abstract) explicit values of their thickness, which are not given or discussed by the authors. It allows distinguishing between zeolite materials with nanometer thick layers, which the authors review in the second paragraph of the Introduction and those that are simply thin platelets with thicknesses of tens of nanometers. There is a fundamental difference between these two cases and the mentioned extensive discussion in the Introduction implies that the reported clinoptilolite materials could be of the first type. However, it appears that the second option is applicable here. If so, then it makes no sense to talk about random lamellae and loss of long-range order as these terms are not used for crystals, even thin. The X-ray diffraction shows that the 3D order of the clinoptilolite is preserved except on a shorter distance (tens of nanometers) because the crystals may be thin.
To summarize, the narrative presented by the authors is confusing and misleading because it adopts the language and terminology that is used to describe two-dimensional material with ultrathin layers, not the ones they present here which are simply think platelets. Adopting the former make these materials appear more attractive but it has to be proven (layer thickness).
Textural properties (Table S1) are poor, which should be mentioned too.
Author Response
Reponse to Revivwer 3
